# OTGM: Graph Matching with Noisy Correspondence via Optimal Transports

## Abstract

Graph matching is a significant task for handling the matching problem of finding correspondences between keypoints in different graphs. Prior research primarily concentrates on performing one-to-one matching in topologic perspective for keypoints across various graphs, assuming that the paired keypoints are accurately linked. However, these approaches have two limitations: (1) because of different observation perspectives, some keypoints in the reference figure may become occluded or transformed, leading to situations where keypoint matches are a mess in topologic; (2) in practice, the manual annotation process is susceptible to poor recognizability and viewpoint differences between images, which probably results in offset and even erroneous keypoint annotations. To address these limitations, we revisit the graph matching problem from the distributional alignment perspective and propose an **O**ptimal **T**ransport **G**raph **M**atching model (**OTGM**). Specifically, (1) to effectively model the real-world keypoint matching scenarios, we have redefined the graph matching process as a transportation plan, which involves transferring node or edge sets from one distribution to another while minimizing the Wasserstein distance between these distributions. (2) To achieve robust matching, we introduce a well-designed graph denoising module to eliminate noisy edges in the input graph with the assistance of self-supervised learning. On top of this, we theoretically provide assurances regarding the generalization ability of OTGM. Furthermore, comprehensive experiments on three real-world datasets demonstrate that our model exhibits strong robustness and achieves state-of-the-art performance compared to competitive baselines.

## 1 Introduction

Graph Matching (GM) Cho et al. (2010); Zanfir & Sminchisescu (2018) is instrumental in establishing correspondences between keypoints across different graphs. This method is crucial in a wide range of applications, including object tracking Yang et al. (2021); Ufer & Ommer (2017), scene graph discovery Chen et al. (2020a), simultaneous localization and mapping (SLAM) Cadena et al. (2016), and structure-from-motion Sarlin et al. (2020b). At the core of GM lies the challenge of unraveling and harnessing bi-level affinities: node-to-node and edge-to-edge matching.

In past decades, recent approaches have concentrated on integrating these dual-levels of information through the design of graph neural networks Wang et al. (2019; 2020b); Sarlin et al. (2020b); Yu et al. (2019) and the implementation of differentiable quadratic losses Gao et al. (2021); Rolínek et al. (2020). Leveraging encoded high-order geometrical data, these innovations in graph matching have led to notable improvements in the precision of correspondence estimation. As shown in Figure 1, although recent progress in GM has shown encouraging results, it is impeded by two fundamental challenges stemming from the basic assumptions and methodologies prevalent in current GM practices as follows:

**C1 Feature-specific Keypoint Matching:** The diversity in observation perspectives often leads to occlusion or overlap of keypoints in reference maps, necessitating the need for semantical-level matches. Conventional GM methods, predominantly based on the topologic-level one-to-one matching paradigm, prove to be inadequate for these real-world scenarios.

**C2 Noisy Annotations in Keypoint Matching:** The manual keypoint annotation process encounters significant difficulties, such as poor recognizability Bourdev & Malik (2009)

and varying viewpoints between images Min et al. (2019). As mentioned by Lin et al. (2023), these issues commonly result in inaccurate keypoint annotations, varying from slight misalignments to entirely incorrect identifications.

Based on the considerations above, this paper focuses on the following question: *Can we find a new GM model, that can conduct the semantic-level matching while mitigating the impact of noise annotations, hence realizing robust matching?*

To address these challenges, we introduce a novel **O**ptimal **T**ransport **G**raph **M**atching (**OTGM**) model, which reformulates the graph matching problem from a distributional alignment perspective. Confronting challenge (**C1**), we leverage the principles of optimal transport Santambrogio (2015) to model the graph matching process as a transportation plan. This approach enables the movement of node or edge sets from one distribution to another and aims to minimize the Wasserstein distance between the distributions, where the optimal transport matrix can represent the semantic relevance among different nodes. Such an approach can support model-level and edge-level matching scenarios from the semantic perspective, offering a more flexible and accurate representation of real-world graph matching. To tackle the (**C2**), OTGM incorporates a robust denoising module designed to filter out noisy edges in the

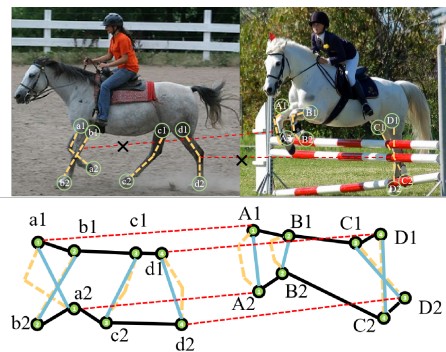

Figure 1: An illustrative example of topologic-level matching with the noisy annotations. Due to occlusion as well as local positional shifts, the topological positions of the nodes can be misplaced and the problem of matching errors can easily occur.

input graph. This module employs self-supervised graph learning techniques to enhance the accuracy and reliability of the matching process, particularly in scenarios involving noisy or inaccurate input data. We further bolster the OTGM method with theoretical guarantees for its generalization ability. This theoretical foundation provides a solid basis for the practical deployment of OTGM in various contexts. Furthermore, experimental results highlight the superiority of OTGM on three real-world datasets.

The main contributions are summarized as three-fold:

- We present a novel problem formulation in graph matching, focusing on semantic-level keypoint correspondences within the context of noisy graph annotations. In this way, we can tackle the lack of topologic mess caused by occluded or transformed.

- Our proposed method, OTGM, innovatively employs distributional alignment principles derived from optimal transport theory. This is further enhanced with a self-supervised denoising module, equipping OTGM to adeptly handle complex graph matching scenarios in the presence of noisy data. Additionally, we provide a comprehensive theoretical analysis of OTGM, including a detailed generalization bound, strengthening its robustness and applicability.

- Empirical experiments demonstrate the effectiveness of OTGM on three well-established benchmarks. Notably, it achieves significant improvements over current state-of-the-art methods, with absolute gains of 1.1% on Pascal VOC and 1.2% on Spair-71k.

## 2 RELATED WORK

### 2.1 DEEP GRAPH MATCHING

Deep Graph Matching (Deep GM) Zanfir & Sminchisescu (2018); Fey et al. (2020) is dedicated to aligning keypoints across different graphs by analyzing node and edge correlations. Existing methods predominantly harness high-order information within graph structures to augment matching accuracy and are broadly classified into two brunches based on their approach to high-order information utilization. The first brunch encompasses network-designed methods Wang et al. (2020c); Yu et al. (2019); Liu et al. (2021a); Jiang et al. (2021), which implicitly integrate high-order information

via GM-customized networks. For instance, PCA Wang et al. (2019) utilizes graph convolutional networks to amalgamate intra-graph and inter-graph structural information. Similarly, NGM Wang et al. (2021) introduces a matching-aware graph convolution approach, incorporating Sinkhorn iteration Cuturi (2013), to enhance the matching process. The second brunch includes loss-designed methods Liu et al. (2021b); Gao et al. (2021); Rolínek et al. (2020), which explicitly learn high-order information through the application of different quadratic loss functions or optimization strategies. QCDGM Gao et al. (2021), for example, adapts the Frank-Wolfe algorithm into a differentiable format for managing quadratic constraints. In addition, BBGM Rolínek et al. (2020) innovates a differentiable combinatorial solver tailored for quadratic assignment problems.

While deep graph matching (GM) methods have shown promising performance, they typically assume faultless and correctly associated node-to-node and edge-to-edge correspondences. However, poor annotations often result in background noise and clutter, making it nearly impossible to achieve perfect correspondences. Existing efforts Wang et al. (2020a); Sarlin et al. (2020b); Qu et al. (2021); Rolínek et al. (2020); Ren et al. (2022); Lin et al. (2023) to achieve robust GM have primarily focused on addressing outliers and adversarial attacks, or acquire knowledge from a pre-trained model, aiming to enhance robustness against outliers and malicious attacks, rather than explicitly addressing the issue of noisy matching in a unified framework. To the best of our knowledge, this study represents the first attempt to specifically tackle the challenge of topologic mess for keypoints matching under the assumption of noisy graphs, paving the way for more accurate and reliable GM in complex, real-world scenarios.

## 2.2 SELF-SUPERVISED GRAPH LEARNING

Although supervised learning has achieved remarkable success in various applications, acquiring a large labeled dataset can be challenging and costly. To address this limitation, self-supervised learning (SSL) has emerged as a promising alternative. Recent advancement in SSL is the utilization of contrastive learning, which incorporates auxiliary training signals generated from different types of graph data, such as heterogeneous graphs Hwang et al. (2020), spatio-temporal graphs Zhang et al. (2023), and molecular graphs Zhang et al. (2021). By employing contrastive learning in SSL, the quality of graph embeddings can be significantly improved, resulting in enhanced performance on various tasks, including node classification and link prediction.

SSL has proven to be effective in learning high-quality representations of graph data. demonstrating significant potential in graph learning. Contrastive SSL Wu et al. (2021) and generative SSL Li et al. (2023) techniques have been utilized in this domain. One example is GFormer Li et al. (2023), which employs a graph autoencoder to reconstruct masked node interactions for data augmentation. This approach generates augmented training data to facilitate the learning of more effective representations of nodes. The integration of self-supervised graph learning techniques has proven beneficial. For instance, S3-Rec Zhou et al. (2020) utilizes a self-attentive neural architecture and employs four auxiliary self-supervised objectives to capture correlations among different types of data. Moreover, C2DSR Cao et al. (2022) introduces a contrastive cross-domain infomax objective, which enhances the correlation between single-domain and cross-domain node representations.

## 2.3 OPTIMAL TRANSPORT

Optimal transport provides a robust method to infer the correspondence between two distributions. For more details, refer to Appendix 3. Recently, optimal transport has garnered significant attention in various computer vision tasks. For instance, Courty et al. (2016) addresses domain adaptation by learning a transportation plan from the source domain to the target domain. Su et al. (2015) employs optimal transport for 3D shape matching and surface registration. Other applications include generative models Arjovsky et al. (2017); Bunne et al. (2019), and graph matching Xu et al. (2019a;c), among others. Moreover, some studies have utilized optimal transport for correspondence problems Liu et al. (2020); Eisenberger et al. (2020); Song et al. (2021); Saleh et al. (2022). However, these studies primarily focus on the matching of nodes, often neglecting edges, which can capture fine-grained semantic matching. To the best of our knowledge, we are the first to address the graph-matching problem by modeling joint coarse- and fine-grained graph matching in the presence of noise within an optimal transport framework.

## 3 PRELIMINARY FOR OPTIMAL TRANSPORT

In our graph matching framework, we utilize two types of distances for optimal transport (OT) Santambrogio (2015) to facilitate the matching process. Specifically, we employ the Wasserstein distance for node matching and the Gromov-Wasserstein distance for edge matching.

**Wasserstein Distance.** The Wasserstein distance (WD) $D_w(\cdot, \cdot)$ defines an optimal transport distance that measures the discrepancy between each pair of samples across the two domains. Specifically, WD serves as a common measure for comparing two distributions, such as two sets of node embeddings as follows.

**Definition 1** *Consider two discrete distributions, denoted as $\mu \in \mathcal{P}(\boldsymbol{X})$ and $\nu \in \mathcal{P}(\boldsymbol{Z})$, where $\mu$ can be expressed as $\mu = \sum_{i=1}^{n} \boldsymbol{u}_i \delta_{\boldsymbol{x}_i}$, and $\nu$ can be expressed as $\nu = \sum_{j=1}^{m} \boldsymbol{v}_j \delta_{\boldsymbol{z}_j}$. Here, $\delta_{\boldsymbol{x}}$ represents the Dirac function centered on $\boldsymbol{x}$. Let $\Pi(\mu, \nu)$ denote the set of all joint distributions $\gamma(\boldsymbol{x}, \boldsymbol{z})$ with marginals $\mu(\boldsymbol{x})$ and $\nu(\boldsymbol{z})$. The weight vectors $\boldsymbol{u} = \{\boldsymbol{u}_i\}_{i=1}^{n} \in \Delta_n$ and $\boldsymbol{v} = \{\boldsymbol{v}_j\}_{j=1}^{m} \in \Delta_m$ belong to the $n$-dimensional and $m$-dimensional simplex, respectively. In other words, both $\mu$ and $\nu$ are probability distributions, satisfying $\sum_{i=1}^{n} \boldsymbol{u}_i = \sum_{j=1}^{m} \boldsymbol{v}_j = 1$. The Wasserstein distance between the two discrete distributions $\mu$, $\nu$ is defined as:*

$$D_w(\mu, \nu) = \inf_{\gamma \in \Pi(\mu, \nu)} \mathbb{E}_{(\boldsymbol{x}, \boldsymbol{z}) \sim \gamma}[\boldsymbol{c}(\boldsymbol{x}, \boldsymbol{z})]$$
$$= \min_{\boldsymbol{T} \in \Pi(\boldsymbol{u}, \boldsymbol{v})} \sum_{i=1}^{n} \sum_{j=1}^{m} \boldsymbol{T}_{ij} \cdot \boldsymbol{c}(\boldsymbol{x}_i, \boldsymbol{z}_j), \tag{1}$$

*where $\Pi(\boldsymbol{u}, \boldsymbol{v}) = \{\boldsymbol{T} \in \mathbb{R}_+^{n \times m} | \boldsymbol{T}\boldsymbol{1}_m = u, \boldsymbol{T}^\top \boldsymbol{1}_n = v\}$, $\boldsymbol{1}_n$ is an $n$-dimensional all-one vector, and $\boldsymbol{c}(\boldsymbol{x}_i, \boldsymbol{z}_j)$ is the cost function measuring the distance between $\boldsymbol{x}_i$ and $\boldsymbol{z}_j$.*

In the field of graph matching, this distance metric serves as a natural choice for node matching. By minimizing the Wasserstein distance, we can effectively align nodes based on their similarity, thereby achieving an accurate and reliable graph matching result.

**Gromov-Wasserstein Distance.** Different from directly calculating distances between two sets of nodes as in the Wasserstein distance, the Gromov-Wasserstein distance (GWD) can be utilized to assess distances between pairs of nodes within each domain and compare these distances to those in the corresponding domain. The GWD, introduced in the works by Peyré et al. (2016) and Chowdhury & Mémoli (2018), is particularly suited for discrete matching scenarios as follows.

**Definition 2** *Keeping the same notation as in Defination 1, the Gromov-Wasserstein distance between $\boldsymbol{u}$ and $\boldsymbol{v}$ is defined as:*

$$D_{gw}(\mu, \nu) = \inf_{\gamma \in \Pi(\mu, \nu)} \mathbb{E}_{(\boldsymbol{x}, \boldsymbol{z}) \sim \gamma, (\boldsymbol{x}', \boldsymbol{z}') \sim \gamma}[\mathcal{C}(\boldsymbol{x}, \boldsymbol{z}, \boldsymbol{x}', \boldsymbol{z}')]$$
$$= \min_{\boldsymbol{T} \in \Pi(\mu, \nu)} \sum_{i,j,i',j'} \hat{\boldsymbol{T}}_{ij} \hat{\boldsymbol{T}}_{i'j'} \mathcal{C}(\boldsymbol{x}_i, \boldsymbol{y}_j, \boldsymbol{x}_{i'}, \boldsymbol{y}_{j'}) \tag{2}$$

*Here, $\mathcal{C}(\cdot)$ denotes the cost function that evaluates the intra-graph structural similarity between two pairs of nodes $(\boldsymbol{x}_i, \boldsymbol{x}_i')$ and $(\boldsymbol{z}_j, \boldsymbol{z}_j')$. Specifically, $\mathcal{C}(\boldsymbol{x}_i, \boldsymbol{z}_j, \boldsymbol{x}_i', \boldsymbol{z}_j') = ||\boldsymbol{c}_1(\boldsymbol{x}_i, \boldsymbol{x}_i') - \boldsymbol{c}_2(\boldsymbol{z}_j, \boldsymbol{z}_j')||$, where $\boldsymbol{c}_i$, with $i \in [1, 2]$, represents functions that evaluate the node similarity within the same graph, such as the cosine similarity.*

## 4 METHODOLOGY

In this section, we introduce a novel Optimal Transport Graph Matching model termed **OTGM** as illustrated in Figure 2. Our approach begins with a clear definition of the problem and subsequently introduces two modules, including optimal transport matching and graph denoising, together facilitating robust matching in complex scenarios.

**Problem Definition.** Given two images with $n$ and $m$ keypoints ($n \leq m$), graph matching aims to establish the node-to-node correspondence between their graphs $\mathcal{G}_A$ and $\mathcal{G}_B$ based on these keypoints. Suppose $\boldsymbol{V}_A \in \mathbb{R}^{n \times d}$ and $\boldsymbol{V}_B \in \mathbb{R}^{m \times d}$ be the feature matrices of keypoints in $\mathcal{G}_A$ and $\mathcal{G}_B$, respectively,

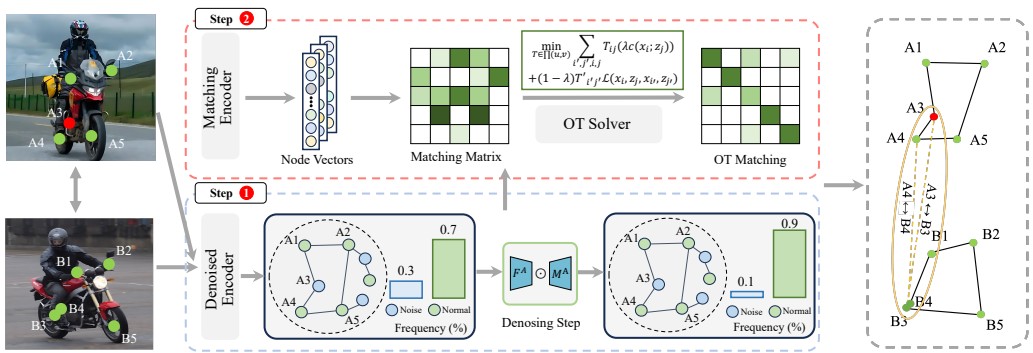

Figure 2: The overall framework of OTGM. In our approach to graph matching, we address the challenge with a dual strategy. Firstly, we implement both node-level and edge-level matching through an optimal transport (OT) module from the semantic perspective, which can facilitate accurate correspondence alignment across graphs. Secondly, we integrate a graph denoising (GD) module, employing self-supervised learning techniques, which significantly enhances the robustness of our method by efficiently filtering out noise and refining the quality of the input graphs. Overall, the synergistic combination of the OT matching module and the GD module culminates in a comprehensive and robust framework, adept at tackling the complexities inherent in graph matching tasks.

and each row of $\boldsymbol{V}_A$ and $\boldsymbol{V}_B$ is a feature vector of a keypoint. Afterward, let $\boldsymbol{F}_A = \boldsymbol{V}_A \boldsymbol{V}_A^T \in \mathbb{R}^{n \times n}$, $\boldsymbol{F}_B = \boldsymbol{V}_B \boldsymbol{V}_B^T \in \mathbb{R}^{m \times m}$ denote the adjacency matrices, embracing the edge associations in graphs $\mathcal{G}_A$ and $\mathcal{G}_B$. Formally, graph matching can be formulated as $\min_Y \mathcal{L}_{\boldsymbol{Y}}(\boldsymbol{Y}_{gt}, \boldsymbol{Y})$, where $\mathcal{L}_Y$ serves to measure the discrepancy between the ground-truth assignment $Y_{\text{gt}}$ and the matching result $Y$, *e.g.*, cross-entropy loss Wang et al. (2021) or hamming distance loss Rolínek et al. (2020). Specifically, it can be formulated as follows:

$$\mathcal{L} = max_{\boldsymbol{Y} \in \Pi} \quad tr(\boldsymbol{S}\boldsymbol{Y}^T) - tr(\boldsymbol{S}\boldsymbol{Y}_{gt}^T) \tag{3}$$

where $\Pi$ represents the set of all $n \times m$ permutation matrices, and $\boldsymbol{S} = \boldsymbol{V}_A \boldsymbol{V}_B^T \in \mathbb{R}^{n \times m}$. By minimizing the objective defined in Eq.(3), the encoder is trained to accurately assign keypoints from one image to another, thereby facilitating effective graph matching.

## 4.1 OPTIMAL TRANSPORT (OT) FOR GRAPH MATCHING

As mentioned earlier, topologic-level graph matching suffers from scenarios where nodes are occluded or transformed. Notice that, there are both feature-invariant semantic and feature-specific semantics among paired nodes in different graphs due to the graph is not completely the same. To this end, we propose semantic-level graph matching, which can learn the similarities information from semantic-relevant nodes and edges.

**Feature-invariant Graph Matching**. Drawing from prior research Liu et al. (2022); Lin et al. (2023), contrastive learning emerges as an efficient and differentiable approach to the linear assignment problem. In our method, we align the keypoints $\boldsymbol{V}_A$ and $\boldsymbol{V}_B$ in accordance with $\boldsymbol{Y}_{gt}$, retaining only those keypoints with corresponding counterparts for training. This process yields aligned keypoints $\boldsymbol{P}_A, \boldsymbol{P}_B \in \mathbb{R}^{n \times d}$ for graphs $\mathcal{G}_A$ and $\mathcal{G}_B$, respectively. We then apply contrastive learning to both the rows and columns of the node similarity matrix $\boldsymbol{S}$, as Radford et al. (2021):

$$\mathcal{L}_{\text{InfoNCE}} = \mathcal{H}(\boldsymbol{I}_n, \rho(\boldsymbol{P}_A \boldsymbol{P}_B^T)) + \mathcal{H}(\boldsymbol{I}_n, \rho(\boldsymbol{P}_B \boldsymbol{P}_A^T)), \tag{4}$$

where $\boldsymbol{I}_n$ is the identity matrix, $\mathcal{H}$ is the row-wise cross-entropy function with mean reduction and $\rho$ is the Softmax function applied row-wise such that each row sums to one,

**Feature-specific Graph Matching**. On the basis of the above, we concentrate on capturing the feature-specific matching scenarios. Let $\boldsymbol{x}$ and $\boldsymbol{z}$ represent keypoints within the feature matrices $\boldsymbol{V}_A \in \mathbb{R}^{m \times d}$ and $\boldsymbol{V}_B \in \mathbb{R}^{n \times d}$ of $\mathcal{G}_A$ and $\mathcal{G}_B$, respectively. We approach the graph matching problem as an Optimal Transport (OT) problem. In this formulation, the transportation cost for moving a unit from keypoint $i$ to keypoint $j$ is denoted as $\boldsymbol{c}_{ij}$. The objective of the OT problem is to derive

an optimal transportation plan $\boldsymbol{\pi}^* = \{\boldsymbol{\pi}_{ij} \mid i = 1, 2, \ldots, m, j = 1, 2, \ldots, n\}$, which allows for the movement of all keypoints $\boldsymbol{x}$ to keypoints $\boldsymbol{z}$ while minimizing the total transport cost:

$$\min_{\boldsymbol{\pi}} \sum_{i=1}^{m} \sum_{j=1}^{n} \boldsymbol{c}_{ij} \boldsymbol{\pi}_{ij} \quad \text{s.t.} \sum_{i=1}^{m} \boldsymbol{\pi}_{ij} = \boldsymbol{z}_j, \ j = 1, \ldots, n, \quad \sum_{j=1}^{n} \boldsymbol{\pi}_{ij} = \boldsymbol{x}_i, \ i = 1, \ldots, m,$$

$$\sum_{i=1}^{m} \boldsymbol{x}_i = \sum_{j=1}^{n} \boldsymbol{z}_j, \quad \boldsymbol{\pi}_{ij} \geq 0, \ i = 1, \ldots, m, \ j = 1, \ldots, n. \tag{5}$$

To model matching at both the node and edge levels, we introduce a shared transport plan $\boldsymbol{T}$, which is utilized in both the Wasserstein Distance (WD) and the Gromov-Wasserstein Distance (GWD). Intuitively, a shared transport plan allows WD and GWD to synergistically enhance each other's effectiveness, as $\boldsymbol{T}$ leverages information from both nodes and edges concurrently. Formally, we define the proposed Optimal Transport (OT) distance as:

$$D_{\text{OT}}(\mu, \nu) = \min_{\boldsymbol{T} \in \Pi(u,v)} \sum_{i',j',i,j} \boldsymbol{T}_{ij} \left( \lambda \boldsymbol{c}(\boldsymbol{x}_i; \boldsymbol{z}_j) + (1 - \lambda) \boldsymbol{T}'_{i'j'} \mathcal{C}(\boldsymbol{x}_i, \boldsymbol{z}_j, \boldsymbol{x}_{i'}, \boldsymbol{z}_{j'}) \right) \tag{6}$$

where $\mathcal{C}(\boldsymbol{x}_i, \boldsymbol{y}_j, \boldsymbol{z}'_i, \boldsymbol{z}'_j) = ||\boldsymbol{c}_1(\boldsymbol{x}_i, \boldsymbol{x}'_i) - \boldsymbol{c}_2(\boldsymbol{z}_j, \boldsymbol{z}'_j)||$, and $\boldsymbol{c}_i, i \in [1, 2]$ are functions that evaluate node similarity (*e.g.*, the cosine similarity).

To obtain a unified solver for the OT distance, we define the unified cost function as:

$$\mathcal{C}_{\text{Unified}} = \lambda \boldsymbol{c}(\boldsymbol{x}, \boldsymbol{z}) + (1 - \lambda) \mathcal{C}(\boldsymbol{x}, \boldsymbol{z}, \boldsymbol{x}', \boldsymbol{z}') \tag{7}$$

where $\lambda$ is the hyper-parameter for controlling the importance of different cost functions. Instead of using projected gradient descent or conjugated gradient descent as in Xu et al. (2019b); Vayer et al. (2019), we can approximate the transport plan $\boldsymbol{T}$ as shown in Algorithm 1 in Appendix A.

In this way, we can conduct the feature-invariant/specific graph matching in a unified framework. Then, the overall OT loss for graph matching is given as follows:

$$\mathcal{L}_{\text{OT}} = \mathcal{L}_{\text{InfoNCE}} + D_{\text{OT}}(\mu, \nu) \tag{8}$$

**Remark 1** *The utilization of these two types of distances, namely the Wasserstein distance for node matching and the Gromov-Wasserstein distance for edge matching, enables us to perform accurate and comprehensive graph matching, taking into consideration both the node and edge characteristics of the graphs.*

## 4.2 GRAPH DENOISING (GD) FOR ROBUST MATCHING

To improve the quality of graph matching, we propose a graph denoising module to filter out noisy information in the input graph. This parameterized network is shown in Figure 2. The main concept behind our approach is to actively filter out noisy edges in the input graph using a parameterized network. For the graph $\mathcal{G}_A$ and $\mathcal{G}_B$, we use the binary matrix $\boldsymbol{M}^{\mathcal{G}}$, *e.g.*, $\boldsymbol{M}^A \in \{0, 1\}^{n \times n}$, $\boldsymbol{M}^B \in \{0, 1\}^{m \times m}$, and $\boldsymbol{M}^{AB} \in \{0, 1\}^{n \times m}$ where $m_{ij}$ denotes whether the edge between node $\boldsymbol{u}_i$ and $\boldsymbol{u}_j$ is present (0 indicates a noisy edge).

Formally, recalling Sec.4.1, the adjacency matrix of the resulting subgraph is $\boldsymbol{F}'_A = \boldsymbol{F}_A \odot \boldsymbol{M}^A$, $\boldsymbol{F}'_B = \boldsymbol{F}_B \odot \boldsymbol{M}^B$, and $\boldsymbol{F}'_S = \boldsymbol{S} \odot \boldsymbol{M}^{AB}$ where $\odot$ is the element-wise product. The straightforward idea to reduce noisy edges with the least assumptions about $\boldsymbol{F}'_A$, $\boldsymbol{F}'_B$ and $\boldsymbol{F}'_S$ is to penalize the number of non-zero entries in $\boldsymbol{M}^{\mathcal{G}}$ of different layers, where $\mathcal{G}$ represents $A$, $B$ or $AB$.

$$\sum_{\mathcal{G}=A,B,AB} \|\boldsymbol{M}^{\mathcal{G}}\|_0 = \sum_{\mathcal{G}=A,B,AB} \sum_{(u,v) \in \mathcal{E}} \mathbf{1}_{\{m_{ij}^{\mathcal{G}} \neq 0\}} \tag{9}$$

where $\mathbf{1}[\cdot]$ is an indicator function, with $\mathbf{1}[\text{True}] = 1$ and $\mathbf{1}[\text{False}] = 0$, $\|\cdot\|_0$ represents the $l_0$ norm. However, because of its combinatorial and non-differentiability nature, optimizing this penalty is computationally intractable. Therefore, we consider each binary number $m_{ij}^{\mathcal{G}}$ to be drawn from a Bernoulli distribution parameterized by $\pi_{ij}^{\mathcal{G}}$, *i.e.*, $\boldsymbol{m}_{ij}^{\mathcal{G}} \sim \text{Bern}\left(\pi_{ij}^{\mathcal{G}}\right)$. Here, $\pi_{ij}^{\mathcal{G}}$ describes the quality of the edge $(u, v)$. To efficiently optimize subgraphs with gradient methods, we adopt

the reparameterization trick and relax the binary entries $m_{ij}^{\mathcal{G}}$ from being drawn from a Bernoulli distribution to a deterministic function $g$ of parameters $\boldsymbol{\alpha}_{ij}^{\mathcal{G}} \in \mathbb{R}$ and an independent random variable $\epsilon^{\mathcal{G}}$. That is $m_{ij}^{\mathcal{G}} = g\left(\boldsymbol{\alpha}_{ij}^{\mathcal{G}}, \epsilon^{\mathcal{G}}\right)$.

Based on the above operations, we design a denoising module to learn the parameter $\boldsymbol{\alpha}_{ij}^{\mathcal{G}}$ that controls whether to remove the edge $(u, v)$. For the graph $\mathcal{G}$, we calculate $\boldsymbol{\alpha}_{ij}^{\mathcal{G}}$ for user node $u$ and its interacted item node $v$ with $\boldsymbol{\alpha}_{i,j}^{\mathcal{G}} = h^{\mathcal{G}}(\epsilon_i^{\mathcal{G}}, \epsilon_j^{\mathcal{G}})$, where $h^{\mathcal{G}}$ is an MLP parameterized by $\theta^{\mathcal{G}}$. In order to get $m_{i,j}^{\mathcal{G}}$, we also utilize the concrete distribution along with a hard Sigmoid function. Within the above formulation, the constraint on the number of non-zero entries in $M^{\mathcal{G}}$ in Eq.(9) can be reformulated as follows:

$$\mathcal{L}_{GD} = \sum_{\mathcal{G}=A,B,AB}^{L} \sum_{(\boldsymbol{u}_i, \boldsymbol{v}_j) \in \mathcal{E}} (1 - \mathbb{P}_{\sigma(\boldsymbol{s}_{i,j}^{\mathcal{G}})}(0|\theta^{\mathcal{G}})), \tag{10}$$

where $\mathbb{P}_{\sigma(\boldsymbol{s}_{i,j}^{\mathcal{G}})}$ denotes the cumulative distribution function (CDF) of $\sigma(\boldsymbol{s}_{i,j}^{\mathcal{G}})$, $\sigma(\cdot)$ is a function that extends the range of $\boldsymbol{s}_{i,j}^{\mathcal{G}}$, and $\boldsymbol{s}_{i,j}^{\mathcal{G}}$ follows a binary concrete distribution, with $\boldsymbol{\alpha}_{i,j}^{\mathcal{G}}$ parameterizing its location.

**Overall Loss**. Finally, combining Eq.(8) and Eq.(10), the overall graph matching loss is formulated as follows,

$$\mathcal{L}_{OTGM} = \beta \mathcal{L}_{OT} + (1 - \beta)\mathcal{L}_{GD} \tag{11}$$

where $\beta$ is the hyper-parameters.

**Remark 2** *Distinct from the majority of existing distillation approaches that derive knowledge from pre-trained models Hinton et al. (2015); Touvron et al. (2021), our method introduces innovative capabilities:*

1. *It generates contrastive views for matching graphs through random node and edge dropout operations. This process facilitates more effective graph matching by optimizing the agreement between the embeddings of these contrastive views.*

2. *It enhances the matching performance by bootstrapping, without the necessity for external knowledge or additional models.*

### 4.3 THEORETICAL ANALYSIS

In the above parts, we have established the OTGM model, here we take a step further and study the generalization ability of our model.

**Notation Definitions**. Without loss of generality, we designate $\mathcal{G}_B$ and $\mathcal{G}_A$ as benchmarks for matching, using $\hat{\mathcal{P}}_A$ and $\hat{\mathcal{P}}_B$ to represent the distributions of $\mathcal{G}_A$ and $\mathcal{G}_B$, respectively. Let $\mathcal{L}$ represent any symmetric loss function that is k-Lipschitz and satisfies the triangle inequality. Let $\phi : \mathbb{R} \to [0,1]$ and a labeling function $f$. A joint distribution $\Pi(\mu_A, \mu_B)$ over $\mu_A$ and $\mu_B$ are $\phi$-Lipschitz transferable if for all $\lambda > 0$, we have $\mathcal{P}_{(\boldsymbol{x}_1, \boldsymbol{x}_2) \sim \Pi(\mu_A, \mu_B)}[|f(\boldsymbol{x}_1) - f(\boldsymbol{x}_2)| > \lambda d(\boldsymbol{x}_1, \boldsymbol{x}_2)] \leq \phi(\lambda)$. Consider $err_B(f) =: \mathbb{E}_{(\boldsymbol{x}, \boldsymbol{y}) \sim \hat{\mathcal{P}}_B}[\mathcal{L}(\boldsymbol{y}, f(\boldsymbol{x}))]$. Define $\Pi^* = \arg\min_{\Pi \in \Pi(\hat{\mathcal{P}}_A, \hat{\mathcal{P}}_B)} \int d(\boldsymbol{x}_A, \boldsymbol{x}_B) + \mathcal{L}(\boldsymbol{y}_A, \boldsymbol{y}_B) d\Pi(\boldsymbol{x}_A, \boldsymbol{y}_A; \boldsymbol{x}_B, \boldsymbol{y}_B)$ and denote $W_1(\hat{\mathcal{P}}_A, \hat{\mathcal{P}}_B)$ as the associated 1-Wasserstein distance. Let $f^* \in \mathcal{H}$ be a Lipschitz labeling function satisfying the $\phi$-probabilistic transfer Lipschitzness (PTL) assumption *w.r.t.* $\Pi^*$, and minimizing the joint error $err_A(f^*) + err_B(f^*)$ *w.r.t* all PTL functions compatible with $\Pi^*$. We assume that the input instances are bounded *s.t.* $|f^*(\boldsymbol{x}_1) - f^*(\boldsymbol{x}_2)| < L_1$ for all $\boldsymbol{x}_1, \boldsymbol{x}_2$.

**Theorem 1** *Consider a sample of $N_A$ labeled instances drawn from $P_A$ and $N_B$ instances to be matched drawn from $\mu_B$, and then for all $\lambda > 0$, with $a = k\lambda$, we have with probability at least $1 - \delta$:*

$$err_B(f) < W_1(\hat{\mathcal{P}}_A, \hat{\mathcal{P}}_B) + \sqrt{\frac{2}{c} \log(\frac{1}{\delta})(\frac{1}{\sqrt{N_A}} + \frac{1}{\sqrt{N_B}})} + err_A(f^*) + err_B(f^*) + kL_1\phi(\lambda).$$

$$\tag{12}$$

*where $N_A$, $N_B$ are the number of nodes in graphs $\mathcal{G}_A$ and $\mathcal{G}_B$, respectively. $c$ is a constant.*

Table 1: Keypoint matching accuracy (%) on Pascal VOC with standard intersection filtering. The best and second-best results are **highlighted** and underlined, respectively.

| Method | Aero | Bike | Bird | Boat | Bottle | Bus | Car | Cat | Chair | Cow | Table | Dog | Horse | Mbike | Person | Plant | Sheep | Sofa | Train | Tv | Mean |
|---|---|---|---|---|---|---|---|---|---|---|---|---|---|---|---|---|---|---|---|---|---|
| GMN | 41.6 | 59.6 | 60.3 | 48.0 | 79.2 | 70.2 | 67.4 | 64.9 | 39.2 | 61.3 | 66.9 | 59.8 | 61.1 | 59.8 | 37.2 | 78.2 | 68.0 | 49.9 | 84.2 | 91.4 | 62.4 |
| PCA | 49.8 | 61.9 | 65.3 | 57.2 | 78.8 | 75.6 | 64.7 | 69.7 | 41.6 | 63.4 | 50.7 | 67.1 | 66.7 | 61.6 | 44.5 | 81.2 | 67.8 | 59.2 | 78.5 | 90.4 | 64.8 |
| NGM | 50.1 | 63.5 | 57.9 | 53.4 | 79.8 | 77.1 | 73.6 | 68.2 | 41.1 | 66.4 | 40.8 | 63.5 | 63.5 | 45.6 |  | 77.1 | 69.3 | 65.5 | 79.2 | 88.2 | 64.1 |
| IPCA | 53.8 | 66.2 | 67.1 | 61.2 | 80.4 | 75.3 | 72.6 | 72.5 | 44.6 | 65.2 | 54.3 | 67.2 | 67.9 | 64.2 | 47.9 | 84.4 | 70.8 | 64.0 | 83.8 | 90.8 | 67.7 |
| LCS | 46.9 | 58.0 | 63.6 | 69.9 | 87.8 | 79.8 | 71.8 | 60.3 | 44.8 | 64.3 | 79.4 | 57.5 | 64.4 | 57.6 | 52.4 | 96.1 | 62.9 | 65.8 | 94.4 | 92.0 | 68.5 |
| CIE | 52.5 | 68.6 | 70.2 | 57.1 | 82.1 | 77.0 | 70.7 | 73.1 | 43.8 | 69.9 | 62.4 | 70.2 | 70.3 | 66.4 | 47.6 | 85.3 | 71.7 | 64.0 | 83.9 | 91.7 | 68.9 |
| QC-DGM | 49.6 | 64.6 | 67.1 | 62.4 | 82.1 | 79.9 | 74.8 | 73.5 | 43.0 | 68.4 | 66.5 | 67.2 | 71.4 | 70.1 | 48.6 | 92.4 | 69.2 | 70.9 | 90.9 | 92.0 | 70.3 |
| DGMC | 50.4 | 67.6 | 70.7 | 70.5 | 87.2 | 85.2 | 82.5 | 74.3 | 46.2 | 69.4 | 69.9 | 73.9 | 73.8 | 65.4 | 51.6 | 98.0 | 73.2 | 69.6 | 94.3 | 89.6 | 73.2 |
| BBGM | 61.9 | 71.1 | 79.7 | 79.0 | 87.4 | 94.0 | 89.5 | 80.2 | 56.8 | 79.1 | 64.6 | 78.9 | 76.2 | 75.1 | 65.2 | 98.2 | 77.3 | 77.0 | 94.9 | 93.9 | 79.0 |
| NGM-v2 | 61.8 | 71.2 | 77.6 | 78.8 | 87.3 | 93.6 | 87.7 | 79.8 | 55.4 | 77.8 | 89.5 | 78.8 | 80.1 | 79.2 | 62.6 | 97.7 | 77.7 | 75.7 | 96.7 | 93.2 | 80.1 |
| SCGM | 62.9 | 72.9 | 79.6 | 79.5 | 89.3 | 94.1 | 89.1 | 79.2 | 58.4 | 79.3 | 80.5 | 79.9 | 79.5 | 76.8 | 64.8 | 98.1 | 78.0 | 75.9 | 98.0 | 93.2 | 80.5 |
| ASAR | 62.9 | 74.3 | 79.5 | 80.1 | 89.2 | 94.0 | 88.9 | 78.9 | 58.8 | 79.8 | 88.2 | 78.9 | 79.5 | 77.9 | 64.9 | 98.2 | 77.5 | 77.1 | 98.6 | 93.7 | 81.1 |
| COMMON | 65.6 | 75.2 | 80.8 | 79.5 | 89.3 | 92.3 | 90.1 | 81.8 | 61.6 | 80.7 | 95.0 | 82.0 | 81.6 | 79.5 | 66.6 | 98.9 | 78.9 | 80.9 | 99.3 | 93.8 | 82.7 |
| CREAM | 67.0 | 75.6 | 82.2 | 78.1 | 89.4 | 91.6 | 89.3 | 81.6 | 62.1 | 82.3 | 74.3 | 81.7 | 80.9 | 79.0 | 67.7 | 99.3 | 78.9 | 73.7 | 98.3 | **94.7** | 81.4 |
| GMTR | **69.0** | 74.2 | 84.1 | 75.9 | 87.7 | 94.2 | 90.9 | **87.8** | 62.7 | 83.5 | 93.9 | 84.0 | 78.7 | 79.6 | 69.2 | 99.3 | **82.5** | **83.0** | 99.1 | 93.3 | 83.6 |
| OTGM | 68.8 | **76.4** | **84.5** | **81.6** | **90.9** | 94.8 | **92.4** | 85.7 | **63.8** | **84.1** | **96.6** | **84.1** | **83.5** | **82.0** | 68.9 | **99.4** | 81.2 | 82.4 | **99.4** | **94.7** | **84.7** |

The detailed proof of Theorem 1 can refer to Appendix B. The term $W_1(\hat{\mathcal{P}}_A, \hat{\mathcal{P}}_B)$ corresponds to the objective function Eq.(7), and in our paper, we minimize the Wasserstein distance between nodes and edges to achieve this goal. The term $\sqrt{\frac{2}{c}\log(\frac{1}{\delta})(\frac{1}{\sqrt{N_A}} + \frac{1}{\sqrt{N_B}})}$ is related with the scale of the datasets. The terms $err_A(f^*)$ and $err_B(f^*)$ respond to the joint error minimizer, illustrating that the property for original graphs, and we utilize the graph denoising to minimize these terms, measuring the noisy degree of the annotation data, and in this paper, we introduce the graph denoising module to realize this point. The term $\phi(\lambda)$ assesses the probability under which the probabilistic Lipschitzness does not hold.

**Remark 3** *Overall, we provide the generalization bound for graph matching under the assumption of noisy annotations, and one can observe that our method can realize good generalization ability based on the OT matching and graph denoising module.*

## 5 EXPERIMENTS

### 5.1 EXPERIMENTAL SETTINGS

**Datasets**. Our experimental evaluation encompasses three widely-used datasets: Pascal VOC with Berkeley annotation Bourdev & Malik (2009), SPair-71K Min et al. (2019), and Willow Object Class Cho et al. (2013). These datasets have been extensively utilized in the field of Graph Matching and provide diverse graph structures and characteristics for evaluation. To ensure comprehensive evaluation, we report both average performance across all categories and per-category performance. By analyzing the per-category results, we can gain insights into the strengths and weaknesses of our proposed method in handling different object categories.

**Implementation Details**. Our method is implemented using PyTorch 1.10.0 and all evaluations are conducted on an Ubuntu 22.04 OS with an NVIDIA RTX 3090 GPU. The encoder network in our implementation consists of an ImageNet-pretrained VGG16 Simonyan & Zisserman (2014) image encoder, a graph neural network called SplineCNN Fey et al. (2018), and a two-layer projection head Chen et al. (2020b). It is noteworthy that OTGM facilitates feature-specific matching. To ensure a fair comparison with other methods, given a graph A, we generate the final prediction by selecting the node in graph B with the highest matching probability for each node in graph A. This approach aligns with conventional feature-invariant matching evaluations while leveraging the strengths of OT for feature-specific correspondences during the matching process. For more details of experiments, please refer to Appendix C.

**Baselines**. In order to assess the performance of our proposed OTGM method, we compare it with 12 popular deep graph matching methods. These methods include GMN Zanfir & Sminchisescu (2018), PCA Wang et al. (2019), NGM Wang et al. (2021), IPCA Wang et al. (2020b), CIE Yu et al. (2019), DGMC Fey et al. (2020), LCS Wang et al. (2020c), BBGM Rolínek et al. (2020), QC-DGM Gao

Table 2: Keypoint matching accuracy (%) on SPair-71k for all classes. The best and second-best results are **highlighted** and underlined, respectively.

| Method | Aero | Bike | Bird | Boat | Bottle | Bus | Car | Cat | Chair | Cow | Dog | Horse | Mbike | Person | Plant | Sheep | Train | Tv | Mean |
|--------|------|------|------|------|--------|------|------|------|-------|------|------|-------|-------|--------|-------|-------|-------|------|------|
| GMN | 59.9 | 51.0 | 74.3 | 46.7 | 63.3 | 75.5 | 69.5 | 64.6 | 57.5 | 73.0 | 58.7 | 59.1 | 63.2 | 51.2 | 86.9 | 57.9 | 70.0 | 92.4 | 65.3 |
| PCA | 64.7 | 45.7 | 78.1 | 51.3 | 63.8 | 72.7 | 61.2 | 62.8 | 62.6 | 68.2 | 59.1 | 61.2 | 64.9 | 57.7 | 87.4 | 60.4 | 72.5 | 92.8 | 66.0 |
| NGM | 66.4 | 52.6 | 77.0 | 49.6 | 67.7 | 78.8 | 67.6 | 68.3 | 59.2 | 73.6 | 63.9 | 60.7 | 70.7 | 60.9 | 87.5 | 63.9 | 79.8 | 91.5 | 68.9 |
| IPCA | 69.0 | 52.9 | 80.4 | 54.3 | 66.5 | 80.0 | 68.5 | 71.4 | 61.4 | 74.8 | 66.3 | 65.1 | 69.6 | 63.9 | 91.1 | 65.4 | 82.9 | 97.5 | 71.2 |
| CIE | 71.5 | 57.1 | 81.7 | 56.7 | 67.9 | 82.5 | 73.4 | 74.5 | 62.6 | 78.0 | 68.7 | 66.3 | 73.7 | 66.0 | 92.5 | 67.2 | 82.3 | 97.5 | 73.3 |
| NGM-v2 | 68.8 | 63.3 | 86.8 | 70.1 | 69.7 | 94.7 | 87.4 | 77.4 | 72.1 | 80.7 | 74.3 | 72.5 | 79.5 | 73.4 | 98.9 | 81.2 | 94.3 | 98.7 | 80.2 |
| BBGM | 75.3 | 65.0 | 87.6 | 78.0 | 69.8 | 94.0 | 87.8 | 78.3 | 72.8 | 82.7 | 76.6 | 76.3 | 80.1 | 75.0 | 98.7 | 85.2 | 96.3 | 98.0 | 82.1 |
| ASAR | 72.4 | 61.8 | 91.8 | 79.1 | 71.2 | 97.4 | 90.4 | 78.3 | **74.2** | 83.1 | 77.3 | 77.0 | 83.1 | 76.4 | 99.5 | 85.2 | 97.8 | 99.5 | 83.1 |
| GMTR | 75.6 | 67.2 | 92.4 | 76.9 | 69.4 | 94.8 | 89.4 | 77.5 | 72.1 | 86.3 | 77.5 | 72.2 | 86.4 | 79.5 | 99.6 | 84.4 | 96.6 | 99.7 | 83.2 |
| COMMON | 77.3 | 68.2 | 92.0 | 79.5 | 70.4 | 97.5 | 91.6 | 82.5 | 72.2 | 88.0 | 80.0 | 74.1 | 83.4 | 82.8 | **99.9** | 84.4 | 98.2 | 99.8 | 84.5 |
| CREAM | 78.4 | **70.3** | 90.5 | 78.6 | **72.1** | 98.5 | 91.7 | 82.0 | 71.4 | 87.1 | **82.4** | 75.4 | 83.5 | **84.4** | 99.4 | **86.0** | 99.5 | 99.9 | 85.1 |
| GMTR | 75.6 | 67.2 | 92.4 | 76.9 | 69.4 | 94.8 | 89.4 | 77.5 | 72.1 | 86.3 | 77.5 | 72.2 | 86.4 | 79.5 | 99.6 | 84.4 | 96.6 | 99.7 | 83.2 |
| OTGM | **78.8** | 69.5 | **93.7** | **81.2** | 72.0 | **98.8** | **92.8** | **83.7** | 74.0 | **89.5** | 81.7 | **77.5** | 84.9 | 83.6 | **99.9** | 86.2 | 98.6 | **99.9** | **85.9** |

et al. (2021), NGM-v2 Wang et al. (2021), SCGM Liu et al. (2022), COMMON Lin et al. (2023), ASAR Ren et al. (2022), GMTR Guo et al. (2024), and CREAM Ma et al. (2024).

## 5.2 RESULTS ON GRAPH MATCHING

**Results on Pascal VOC**. Pascal VOC is a widely used dataset for object recognition tasks, consisting of 7,020 training images and 1,682 testing images. The dataset contains 20 different object classes, and the number of nodes per graph varies from 6 to 23. In line with the methodology followed in the BBGM approach Rolínek et al. (2020), we preprocess the data by filtering out non-matched points before performing the matching process. This preprocessing step helps to focus on relevant and meaningful correspondences. Table 1 presents the keypoint matching accuracy results on the Pascal VOC dataset. Our proposed method, OTGM, achieves superior performance compared to the other methods, with an improvement of +1.1% in terms of accuracy. Notably, our method demonstrates remarkable performance improvements in classes with challenging and noisy annotations, such as "aero" with +1.3% improvement and "car" with +1.4% improvement.

**Results on Willow Object**. Willow Object is a dataset that contains 256 images distributed across 5 categories. Each target object in the dataset is annotated with 10 distinctive landmarks, providing valuable information for keypoint matching tasks. To ensure a consistent evaluation, we follow the protocol outlined in PCA, IPCA, and NGM Wang et al. (2019; 2020b; 2021). Specifically, we train our methods using the first 20 images of the dataset and report the testing results on the remaining images. Table 3 presents the keypoint matching accuracy results across all objects in the Willow Object dataset. Our proposed method demonstrates significant improvements over the baseline methods, with an increase in accuracy of +0.5%.

Table 3: Keypoint matching accuracy (%) across all objects on Willow Object.

| Method | Car | Duck | Face | Mbike | Wbottle | Mean |
|--------|------|------|------|-------|---------|------|
| GMN | 67.9 | 76.7 | 99.8 | 69.2 | 83.1 | 79.3 |
| NGM | 84.2 | 77.6 | 99.4 | 76.8 | 88.3 | 85.3 |
| PCA | 87.6 | 83.6 | **100** | 77.6 | 88.4 | 87.4 |
| CIE | 85.8 | 82.1 | 99.9 | 88.4 | 88.7 | 89.0 |
| IPCA | 90.4 | 88.6 | **100** | 83.0 | 88.3 | 90.1 |
| SCGM | 91.3 | 73.0 | **100** | 95.6 | 96.6 | 91.3 |
| ASAR | 92.5 | 84.0 | **100** | 95.4 | 99.0 | 94.2 |
| LCS | 91.2 | 86.2 | **100** | 99.4 | 97.9 | 94.9 |
| DGMC | 98.3 | 90.2 | **100** | 98.5 | 98.1 | 97.0 |
| BBGM | 96.8 | 89.9 | **100** | 99.8 | 99.4 | 97.2 |
| NGM-v2 | 97.4 | 93.4 | **100** | 98.6 | 98.3 | 97.5 |
| QC-DGM | 98.0 | 92.8 | **100** | 98.8 | 99.0 | 97.7 |
| COMMON | 97.6 | 98.2 | **100** | **100** | 99.6 | 99.1 |
| CREAM | 97.7 | 98.4 | **100** | **100** | 99.6 | 99.2 |
| GMTR | 97.5 | 97.8 | **100** | **100** | 99.2 | 99.0 |
| OTGM | **98.8** | **99.1** | **100** | **100** | **99.8** | **99.6** |

**Results on SPair-71k**. SPair-71k is a dataset that consists of 70,958 image pairs collected from Pascal VOC 2012 and Pascal 3D+. In line with the data preparation methods used in PCA, IPCA, and NGM Wang et al. (2019; 2020b; 2021), each object in the dataset is cropped to its bounding box and scaled to a fixed size of $256 \times 256$ pixels. Table 2 presents the keypoint matching accuracy results on the SPair-71k dataset. Our proposed method consistently improves the matching performance by +1.2% compared to the other methods. These results demonstrate the effectiveness of our method

Figure 3: The qualitative visualization.

in enhancing the accuracy of keypoint matching tasks on the SPair-71k dataset. Our approach outperforms existing methods and showcases the potential for improved performance in challenging scenarios.

**Qualitative Results and Visualizations**. In Figure 3, our qualitative visualizations demonstrate the effectiveness of our method in modeling both feature-specific and feature-invariant matching. The figure showcases various examples where our approach successfully matches corresponding nodes and edges between different graphs, highlighting its robustness and flexibility. The left side of the figure depicts matching between different types of buses, illustrating how our method can accurately align corresponding features despite variations in appearance and structure. This example emphasizes the capability of our method to handle complex and heterogeneous data, providing precise node-level and edge-level correspondences. The right side of the figure presents matching between bicycles, further showcasing the versatility of our approach in dealing with objects that have multiple corresponding parts. The visualizations reveal that our method can effectively establish feature-specific correspondences, such as matching different parts of the bicycle frame and wheels, which traditional feature-invariant matching methods might miss.

**Ablation Studies and Parameter Analysis**. To evaluate the effectiveness of our framework, we conduct comprehensive ablation studies where we investigate each component separately. The results, as shown in Figure 4, demonstrate that all modules are integral to our approach and contribute significantly to performance gains. To further analyze the sensitivity of our method to parameter choices, we perform a parameter sensitivity analysis on the $\beta$, as presented in Table 5 in Appendix. The results confirm that our method is found to be relatively insensitive to the choice of $\beta$. These ablation studies provide valuable insights into the effectiveness and robustness of each component of our method. By demonstrating their individual contributions and parameter insensitivity, we establish the efficacy of our framework in addressing noisy annotations and achieving superior matching performance.

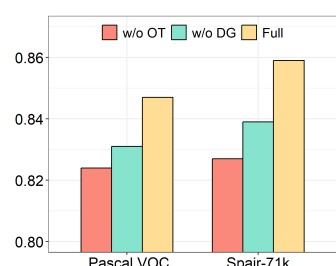

Figure 4: Ablation study of OTGM on Pascal VOC and Spair-71k datasets.

## 6 CONCLUSION

In this study, we introduce Optimal Transport Graph Matching (OTGM), a novel approach designed to address the inherent challenges in graph matching. OTGM redefines graph matching as a distributional alignment problem, effectively addressing errors resulting from viewpoint discrepancies and occlusions, common challenges in graph matching. Additionally, we incorporate a graph denoising module leveraging self-supervised learning techniques, significantly enhancing the robustness of our method by filtering out noise and refining input graph quality. Theoretical analysis within this study substantiates OTGM's strong generalization capabilities. Moreover, comprehensive empirical evaluations across various real-world datasets have demonstrated our method's superiority, outperforming leading baseline models in terms of robustness and overall performance.

**Limitations and Future Work:** OTGM's adaptability to large-scale graphs remains an area for enhancement, as the complexity may impact processing times. Future efforts could explore algorithmic optimizations to better manage large graph datasets. Broader Impacts can refer to Appendix D.

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
