# OpenReview forum: "OTGM: Graph Matching with Noisy Correspondence via Optimal Transports"
_ICLR.cc/2025/Conference — ICLR 2025 Conference Withdrawn Submission_

### Official Review · Reviewer_5hrm · 2024-10-28

**Soundness:** 3
**Presentation:** 3
**Contribution:** 3
**Rating:** 6
**Confidence:** 4

**Summary:**

The paper presents a robust and innovative framework for addressing graph matching under noisy correspondence conditions using optimal transport and a graph denoising module. It provides strong theoretical guarantees and shows improvements on multiple benchmarks.

**Strengths:**

The graph denoising module is a significant contribution. By incorporating self-supervised learning, this module effectively filters noisy edges, which is crucial for handling real-world data with inherent noise or occlusions. This makes the model highly adaptable to imperfect data.

**Weaknesses:**

- The scalability of the model to large-scale graph datasets is not fully addressed. The complexity of optimal transport computations, especially when combined with denoising, could be a bottleneck for real-time or large-scale applications.
- This work focuses on keypoint matching but does not explore other potential applications of graph matching, such as scene graph generation or 3D pose estimation, which are also significant in this field. This could limit the generalization claims of the proposed model.
- Although the authors claim that the denoising module is effective, it inevitably adds considerable complexity to the system. It relies on a binary sampling mechanism, which could increase training time.
- Although the model shows improvements, the gains are relatively incremental on some benchmarks, given the complexity introduced by the model.

**Questions:**

- Is there any efficiency comparison about the system? Further simplification or efficiency improvements in this module might be necessary.
- Could the proposed method be adapted for use in other graph-based tasks? The authors may support with further experiments.

---

### Official Review · Reviewer_KwER · 2024-11-02

**Soundness:** 3
**Presentation:** 3
**Contribution:** 3
**Rating:** 6
**Confidence:** 4

**Summary:**

The paper introduces a novel graph matching model named OTGM (Optimal Transport Graph Matching) to address the challenges of keypoint matching under varying viewpoints and noisy annotations. Its contributions mainly Includes (1) It revisits the graph matching problem from a distributional alignment perspective, redefining the graph matching process as a transportation plan that minimizes the Wasserstein distance between distributions to transfer node or edge sets. (2) It proposes a graph denoising module using self-supervised learning techniques to achieve robust matching. Additionally, it also provides theoretical guarantees on the generalization ability of OTGM. Comprehensive experiments on three real-world datasets and ablation study demonstrates the effectiveness of the proposed modules.

**Strengths:**

1. It employs distributional alignment principles derived from optimal transport theory, and provides theoretical guarantees on the method’s generalization ability.
2. The proposed graph denoising module enhances the matching performance by bootstrapping without the necessity for external knowledge or additional models.
3. Experimental results show the effectiveness of the proposed method, and the ablation study looks reasonable.
4. The supplementary materials and code are provided.

**Weaknesses:**

1. The two innovations in the paper do not seem to be strongly related, which weakens the persuasiveness of the paper as a whole.
2. The paper mainly reflects the effectiveness of the proposed method through the final experimental results. However, the paper’s aims, which is to mitigate the impact of viewpoints and noisy annotations on graph matching, is not easily discernible. It may be necessary to include more experiments, such as those related to noisy correspondences in [1] (Figure 3(a)), and provide more visual examples.

[1] Lin Y, Yang M, Yu J, et al. Graph matching with bi-level noisy correspondence[C]//Proceedings of the IEEE/CVF international conference on computer vision. 2023: 23362-23371.

**Questions:**

1. In Figure 2 step 1, there seems to be no change in the graph before and after denoising, is it a mistake or done on purpose?
2. In line 080-081, “particularly in scenarios involving noisy or inaccurate input data.” Is there experimental evidence or examples to support this claim? I noticed Table. 4 of appendix, but it is a combined result.
3. In line 106, 107, 111, should the ‘brunch’ be ‘branch’?
4. In line 293, how is the value of \lambda determined, what are the results of different selections for \lambda, and have you considered the conditions where \lambda is set to 0 or 1 (only one loss is preserved)?
5. I noticed that the graph denoising is mainly about denoising noise edges, did you try denoising keypoints, like LightGlue[1]?

[1] Lindenberger P, Sarlin P E, Pollefeys M. Lightglue: Local feature matching at light speed[C]//Proceedings of the IEEE/CVF International Conference on Computer Vision. 2023: 17627-17638.

---

### Official Review · Reviewer_Lpc1 · 2024-11-02

**Soundness:** 3
**Presentation:** 3
**Contribution:** 3
**Rating:** 5
**Confidence:** 4

**Summary:**

This paper revisits the graph matching problem from the distributional alignment perspective and proposes an Optimal Transport Graph Matching model called OTGM.   The authors formulate the graph matching process as a transportation plan and  introduce a well-designed graph denoising module to eliminate noisy edges.

**Strengths:**

1. Paper writing, clear representation and good illustration.
2. Sufficient experimental evalutation on the all graph matching benchmarks.

**Weaknesses:**

1. Using optimal transfer for graph matching is not a novel approach. Some previous works have exploited related works, such as [1,2,3,4].  In addition, the role of optimal transfer for semantic-level alignment needs to be further proven. \
[1] Graph matching via optimal transport, arxiv \
[2] Gromov-wasserstein learning for graph matching and node embedding, icml \
[3] DHOT-GM: Robust Graph Matching Using A Differentiable Hierarchical Optimal Transport Framework, arxiv \
[4] Subgraph matching via partial optimal transport,IEEE International Symposium on Information Theory \

2. The verification of the experimental part is too simple. Regarding the noise correspondence in the graph matching task, there is a lack of comparative experiments to solve the noise correspondence. This cannot be explained by improving the accuracy index alone.  More indepth experimental analysis  or theorical discussion about robustness is missing.

3. In the experimental section, the rubustness discussion about graph matching is missing. Figure 3 shows the qualitative visualization. However, the noisy matching is not shown.

**Questions:**

The computation complexity about the new model is needed.

---

### Official Review · Reviewer_FmW2 · 2024-11-05

**Soundness:** 2
**Presentation:** 2
**Contribution:** 2
**Rating:** 5
**Confidence:** 4

**Summary:**

The paper addresses the task of graph matching, which establishes correspondences between keypoints across different graphs. It tackles two major challenges: feature-specific keypoint matching in scenarios with occlusions and the issue of noisy annotations in keypoint matching. To overcome these challenges, the authors introduce a novel Optimal Transport Graph Matching (OTGM) model that reformulates graph matching from a distributional alignment perspective using optimal transport principles and incorporates a robust denoising module. This approach leverages self-supervised graph learning to enhance matching accuracy and provides theoretical guarantees for its generalization ability. Empirical experiments on three real-world datasets demonstrate that OTGM significantly outperforms current state-of-the-art methods, highlighting its effectiveness and robustness.

**Strengths:**

- Traditional methods focus on topological or geometric matches and struggle with occlusions and transformations. By targeting semantic-level correspondences, the approach enhances robustness in complex real-world scenarios.
- Leveraging optimal transport theory allows for more flexible graph alignment, while the self-supervised denoising module effectively handles noisy annotations. This combination improves accuracy and reliability compared to previous fixed matching strategies.

**Weaknesses:**

- The idea of using the distances between correspondences to measure their similarity as the cost function has been used in [ref1].
- The paper writing is kind of obscure.  There are obstacles hindering readers from following the key components of the methods, e.g.
    - There are too many variable names in the paper. This hinders the legibility of the method.
    - Some definitions of variables are not consistent. V_A, V_B in L215 and V_A, V_B in L268.
    - In L287, y_j should not be in the equation.
    - The definition of function c in L185 is c(x_i, z_j). Does the c(;) in Eq. 6 have the same meaning?
    - There are missing definitions of variables, e.g. T'_{i',j'} in Eq. 6, c_{ij} and \pi_{ij} in Eq. 5 are coordinates or some else, L_Y in L240, etc.

[ref1] Zhang X, Yang J, Zhang S, et al. 3D registration with maximal cliques[C]//Proceedings of the IEEE/CVF Conference on Computer Vision and Pattern Recognition. 2023: 17745-17754.

**Questions:**

How did the name of feature-invariant graph matching come about? The invariance of features means the features are fixed. In my opinion, the fixed parts are the matched keypoints. Please explain the meaning of the feature invariance.

---

### Note · Authors · 2025-01-28

I have read and agree with the venue's withdrawal policy on behalf of myself and my co-authors.